# An intervention to improve teacher well-being support and training to support students in UK high schools (the WISE study): A cluster randomised controlled trial

Judi Kidger[1]*, Nicholas Turner[1], William Hollingworth[1], Rhiannon Evans[2], Sarah Bell[1], Rowan Brockman[1], Lauren Copeland[2], Harriet Fisher[1], Sarah Harding[3], Jillian Powell[3], Ricardo Araya[4], Rona Campbell[1], Tamsin Ford[5], David Gunnell[1], Simon Murphy[2], Richard Morris[1]

1 Population Health Sciences, University of Bristol, Bristol, United Kingdom, 2 DECIPHer, School of Social Sciences, Cardiff University, Cardiff, United Kingdom, 3 School for Policy Studies, University of Bristol, Bristol, United Kingdom, 4 Centre for Population Neuroscience and Precision Medicine, King's College London, London, United Kingdom, 5 Department of Psychiatry, University of Cambridge, Cambridge, United Kingdom

* judi.kidger@bristol.ac.uk

**Data Availability Statement:** Data are available at the University of Bristol data repository, data.bris,

## Abstract

### Background

Teachers are at heightened risk of poor mental health and well-being, which is likely to impact on the support they provide to students, and student outcomes. We conducted a cluster randomised controlled trial, to test whether an intervention to improve mental health support and training for high school teachers led to improved mental health and well-being for teachers and students, compared to usual practice. We also conducted a cost evaluation of the intervention.

### Methods and findings

The intervention comprised (i) Mental Health First Aid training for teachers to support students; (ii) a mental health awareness session; and (iii) a confidential staff peer support service. In total 25 mainstream, non-fee-paying secondary schools stratified by geographical area and free school meal entitlement were randomly allocated to intervention ($n = 12$) or control group ($n = 13$) after collection of baseline measures. We analysed data using mixed-effects repeated measures models in the intention-to-treat population, adjusted for stratification variables, sex, and years of experience. The primary outcome was teacher well-being (Warwick-Edinburgh Mental Well-being Scale). Secondary outcomes were teacher depression, absence, and presenteeism, and student well-being, mental health difficulties, attendance, and attainment. Follow-up was at months 12 (T1) and 24 (T2). We collected process data to test the logic model underpinning the intervention, to aid interpretation of the findings. A total of 1,722 teachers were included in the primary analysis. Teacher well-being did not differ between groups at T2 (intervention mean well-being score 47.5, control group

at https://doi.org/10.5523/bris.
2bfr2zmy03bio2wslt3673ey9n.

**Funding:** This study was funded by the National Institute for Health Research's Public Health Research programme 13/164/06 (JK, WH, RE, RA, RC, TF, DG, SM) https://www.nihr.ac.uk/. The intervention costs were covered by Public Health England (JK, WH, RE, RA, RC, TF, DG, SM), Public Health Wales (JK, WH, RE, RA, RC, TF, DG, SM) and Bristol City Council (JK, WH, RE, RA, RC, TF, DG, SM). The funders had no role in study design, data collection and analysis, decision to publish, or preparation of the manuscript.

**Competing interests:** The authors have declared that no competing interests exist.

**Abbreviations:** CACE, Complier Average Causal Effect; CAMHS, Child and Adolescent Mental Health Services; EAP, Employee Assistance Programme; FSM, free school meal; HSC, Healthy Schools Coordinator; ICC, intracluster coefficient; MAR, missing at random; MHFA, Mental Health First Aid; MNAR, missing not at random; PHQ, 8-item Patient Health Questionnaire; RCT, randomised controlled trial; SDQ, Strengths and Difficulties Questionnaire; WEMWBS, Warwick-Edinburgh Mental Well-being Scale; WISE, Well-being in Secondary Education; WPAI, Work Productivity and Activity Impairment.

mean well-being score 48.4, adjusted mean difference −0.90, 95% CI −2.07 to 0.27, $p = 0.130$). The only effect on secondary outcomes was higher teacher-reported absence among the intervention group at T2 (intervention group median number of days absent 0, control group median number of days absent 0, ratio of geometric means 1.04, 95% CI 1.00 to 1.09, $p = 0.042$). Process measures indicated little change in perceived mental health support, quality of relationships, and work-related stress. The average cost of the intervention was £9,103 per school. The study's main limitations were a lack of blinding of research participants and the self-report nature of the outcome measures.

## Conclusions

In this study, we observed no improvements to teacher or student mental health following the intervention, possibly due to a lack of impact on key drivers of poor mental health within the school environment. Future research should focus on structural and cultural changes to the school environment, which may be more effective at improving teacher and student mental health and well-being.

## Trial registration

www.isrctn.com ISRCTN95909211.

---

## Author summary

### Why was this study done?

- Teachers have poorer mental health and well-being than other professions.

- Poor mental health among teachers has a negative impact on student mental health, possibly through poor quality relationships.

- Mental health interventions have been introduced and evaluated in schools, but none have focused on teachers' own mental health.

### What did the researchers do and find?

- We designed and tested an intervention in UK high schools that provided mental health training and a peer support service for teachers.

- We recruited 25 schools and randomly assigned them either to a group that received the intervention or to a comparison group who did not receive any additional mental health support.

- Two years after the intervention, we did not see any differences in mental health and well-being among teachers or students at the intervention schools compared to the comparison schools. Teachers in the intervention group had an average well-being score of 47.5 out of 70, and those in the comparison group had an average score of 48.4 out of 70.

**What do these findings mean?**

- The intervention that we introduced into the schools did not have an impact on teacher or student mental health and well-being.

- This may be because the intervention did not successfully increase how supported teachers and students felt, or improve the quality of relationships.

- It is important to keep trying to find interventions that improve teacher and student mental health. Possibly interventions that have a bigger impact on school culture—including the quality of relationships and level of perceived support—would be more effective.

## Introduction

Teachers are at risk of poor mental health and well-being [1,2]. Causes of work-related stress include excessive workload, challenging student behaviour, and pressure to meet an increasing number of externally determined targets [1,3]. Failure to support teachers may lead to longer-term mental health problems, presenteeism (going to work when one is mentally or physically ill), and sickness absence or quitting the profession [1,4]. Further, teachers' mental health and well-being is associated with student mental health and well-being [5]. Supportive relationships with school staff can be protective of future depression for students in secondary schools [6], whereas poor teacher–student relationships among this age group predict mental health problems and future exclusion [7].

Most school mental health intervention studies have focused on classroom based psychological or educational approaches, with mixed evidence of effectiveness [8]. There is some evidence that whole-school approaches to improving student social, emotional, and/or behavioural adjustment, combining curriculum content, changes to the whole school ethos, and parental involvement, are successful [9]. Far fewer studies have focused on teachers' skills in supporting vulnerable students, although evidence from 2 recent United Kingdom cluster randomised controlled trials (RCTs) has shown that enhancing teacher–student support can improve student outcomes [10,11]. To our knowledge, no school-based trials to date have targeted teacher mental health or well-being as the primary outcome. One systematic literature review examined 4 studies that aimed to support teachers' mental health [12]. Results indicated that changes to the school environment including provision of mentoring could have a positive effect, but quality of the evidence was poor.

Given the evidence of the importance of better support for teachers, both for their own and their students' mental health outcomes, we developed the WISE (Well-being in Secondary Education) intervention, which aimed to improve the support available to teachers for their own mental health and well-being and to increase their skills in supporting students [13]. The intervention makes use of Mental Health First Aid (MHFA) [14], and MHFA for Schools and Colleges, which is a version developed specifically in England for staff in schools. MHFA has been shown to improve social support, and increase lay people's knowledge and confidence in helping others in crisis, and has high participant acceptability [14–17]. However, a recent review found a lack of evidence for impact of MHFA training on mental health outcomes [18]. Drawing on social support theory, we hypothesised that improving workplace social support would have a positive impact on teacher well-being and mental health [19,20]. We further

hypothesised, based on previous findings that teachers' own well-being may affect their abilities to support students both emotionally and academically [21,22], that improvements to teachers' own mental health and well-being should lead to more positive teacher–student relationships, which is associated with improved student mental health [5].

We tested the acceptability and feasibility of the intervention in a pilot study of 6 schools, through observations of the training, interviews/focus groups with training attendees and the wider staff body, and surveys of staff and students [13]. This pilot study found that schools and staff within schools were willing and able to participate in the intervention, that the MHFA training was perceived to be appropriate for the British secondary school context, that it improved mental health knowledge and attitudes, and that the peer support service was feasible to establish and perceived to be helpful. Barriers to the success of the peer support service were identified, which we addressed in this main study. Specifically, we strengthened the guidance provided to peer support services relating to advertising and confidentiality procedures, we increased the number of peer supporters in each school, we extended the length of intervention time by 1 year, and we emphasised the importance of senior leadership support. We also added shorter training component delivered to the whole teaching team, which included information about the peer support services.

We undertook a cluster RCT of the WISE intervention [20] to evaluate its effectiveness at improving teacher and student well-being and mental health, reducing teacher absence, presenteeism, and quitting the profession, and improving student attendance and academic performance, compared to usual practice and the costs associated with this. The trial included an embedded process evaluation [23], which explored how well the intervention was implemented and used in schools (reported elsewhere) [24], and the extent to which the key mechanisms of change hypothesised in the logic model (Fig 1)—increased access to support for staff, improved staff–student relationships, improved mental health support for students, and reduced stress relating to supporting students—were achieved. This paper includes findings relating to this latter process question, reporting on the second and fourth process outcomes shown in the logic model. Forthcoming findings will report on the qualitative process data, which explored perceptions of the intervention in more depth and which, along with the implementation paper [24], relate to the first and third process outcomes.

## Methods

### Study design and participants

Twenty-five mainstream secondary schools (regular schools for students aged 11 to 18 where individuals with and without special educational needs are taught together) were recruited in 2 geographical areas—a city in South West England and surrounding area (comprising a sampling frame of 64 schools) and Central South Wales/South East Wales (comprising a sampling frame of 88 schools). These areas were selected due to proximity to the research team but provided a wide range of schools in terms of academic attainment, inspectorate rating, and socioeconomic catchment areas. Schools were excluded if they were fee paying, alternative provision (that is provision for students unable to attend mainstream school, for example, due to learning or physical disability or because they have been excluded from mainstream education), WISE pilot study schools, participating in similar research studies, already delivering MHFA or other mental health training, or lacking data on proportion of students eligible for free school meals (FSMs) as a proxy measure for poverty. We also excluded schools within the same academy trust (governing body) and local authority as one that had already been recruited, to avoid risk of contamination.

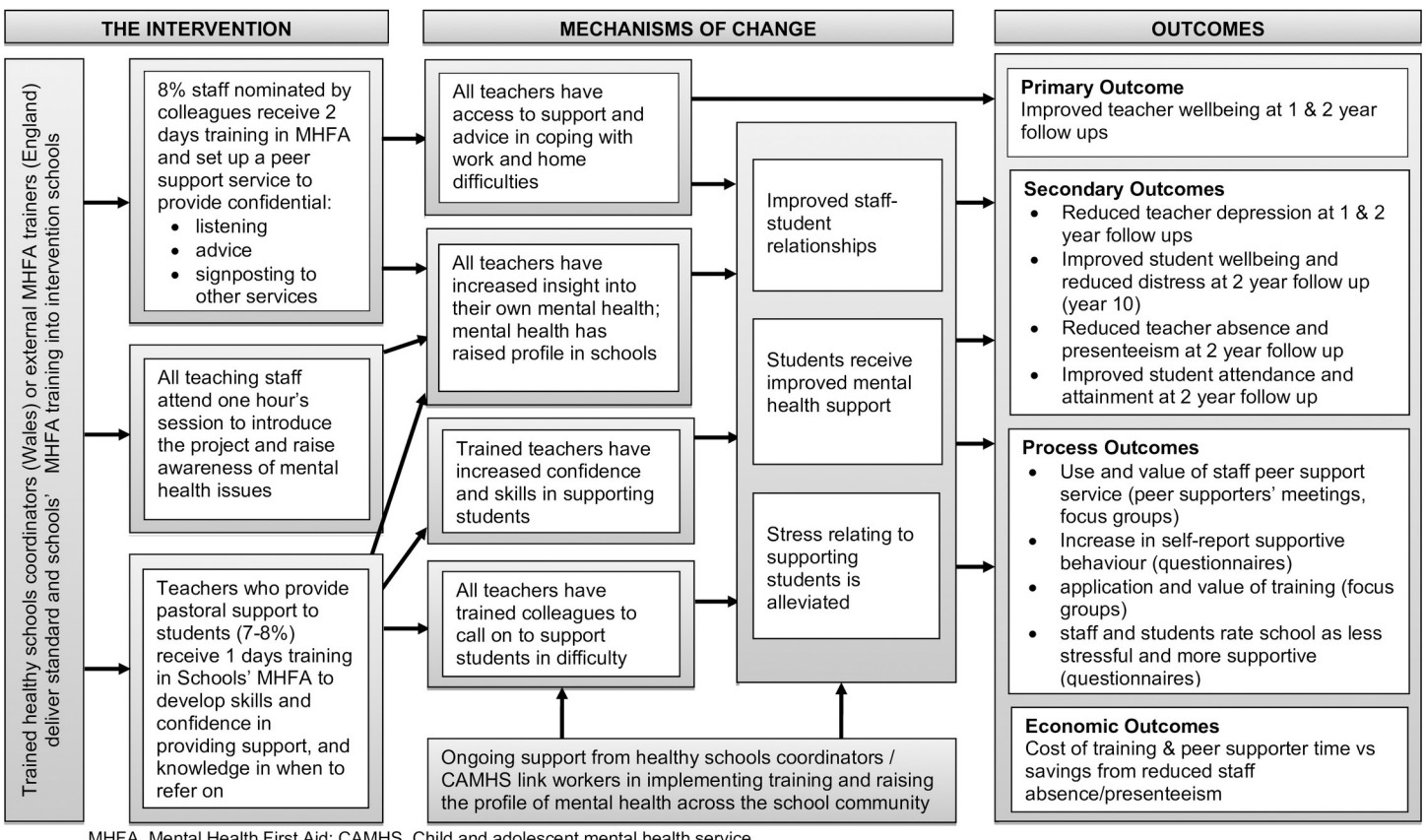

**Fig 1. Logic model of the WISE intervention.** CAMHS, Child and Adolescent Mental Health Services; MHFA, Mental Health First Aid; WISE, Well-being in Secondary Education.

Our study population within each school was all teaching staff and all students in year 8 (aged 12 to 13 years) at baseline. We assumed that any effect on this year group would be representative of effects for all students, as the intervention was delivered at the school level. There were no inclusion or exclusion criteria for teachers or students in year 8. At the follow-up time points, a small number were ineligible to receive the questionnaire, due to having left the school, or being absent from school long term (for example, due to long-term sickness).

Signed research agreements consenting school participation in the study were obtained from all head teachers or their representatives. Teacher and student participants were given information sheets outlining how to opt out of participation—instructions stated that those who did not wish to complete the questionnaire should place the blank copy into the envelope provided and return to the study team. Consent was then assumed for those completing questionnaires. Parents/carers of students were given the option to withdraw their child from all research activities.

No serious adverse events were recorded during any of the data collections. Ethical approval was granted by the University of Bristol's Faculty of Health Sciences Research Ethics Committee (FREC reference number: 28522).

## Randomisation and masking

Schools were the unit of allocation as the intervention operated at the whole-school level. In Wales, the sample was divided into the 2 administrative regions and further stratified into 3

FSM eligibility levels (high, medium, and low proportion of students eligible for FSMs compared to the national average). Two schools were randomly selected from each stratum in each region and invited to participate. Schools that declined were replaced by a randomly selected school from the same stratum and region. In England, schools that were interested in participating following advertisement to all head teachers were stratified according to city/non-city location and high, medium, and low levels of FSM; 2 schools within each location and stratum were then randomly selected for participation. Following discussion about the possibility of school dropout, we randomly selected one additional school for participation from the English sampling frame. In total, there were 12 participating schools in Wales and 13 participating schools in England.

This sampling procedure resulted in 2 recruited schools within each location (England city/England non-city/South Central Wales/South East Wales) and FSM stratum (low/medium/high) combination. Within each of these pairings, one school was randomly allocated to each study arm. Block randomisation was undertaken by the study statistician, who was blinded to the schools' identities, using the Stata *ralloc* command. This process resulted in 12 intervention schools (6 in Wales and 6 in England) and 13 comparison schools (6 in Wales and 7 in England). Random allocation to study arm took place after baseline data collection.

Schools and the study team including those who collected the questionnaire data were not blind to allocation at the follow-up data collections. In the case of school staff, this was because it was not possible to conceal the intervention delivery from them. In the case of the data collections, these staff had also been the ones to organise the intervention delivery due to resource limitations.

## Procedures

All data collections were conducted in school, led by a study team member, accompanied by trained fieldworkers. Individual-level teacher outcomes were collected through self-report paper questionnaires during staff meetings in May to July 2016 (T0, preintervention), May to July 2017 (T1, 1-year follow-up), and May to July 2018 (T2, 2-year follow-up). Those who were absent were offered the option of completion via an online survey. Individual-level student outcomes were gathered via self-report paper questionnaires at T0 and T2 during tutor groups or lesson time. Where less than 5 students in a tutor group were absent, the paper questionnaires were left for them to complete when next in school, under the supervision of school staff. Where more than 5 were absent, a second data collection session administered by a study team member was arranged. At least one teacher was present in the room during student data collections to adhere to school child protection policies and assist with behaviour management. They were given a written information sheet explaining the data collection process and requesting that they did not look at the students' questionnaires or involve themselves with the data collections.

Data on school-level teacher and student outcomes were collected from schools at T0 and T2.

Each intervention school received the following: (i) MHFA training for 8% staff who then provided a confidential peer support service for colleagues; (ii) MHFA for Schools and Colleges training for a further 8% teachers; and (iii) a 1-hour mental health awareness session for staff. In English schools, these sessions were delivered by an independent MHFA trainer. In Welsh schools, Healthy Schools Coordinators (HSCs) who are employed by Public Health Wales to support schools in health improvement work were trained to deliver the sessions. We planned for the HSCs and school link workers based within Child and Adolescent Mental Health Services (CAMHS) to provide ongoing support to the peer support services once they

**i) 2 day standard Mental Health First Aid (MHFA) training and development of staff peer support service**
*Who*
8% staff (teaching and non-teaching) nominated by colleagues at baseline
*Content of training*
 -Signs and symptoms of common mental disorders
- Supporting others using ALGEE steps (Assess for risk, Listen non-judgmentally, Give information, Encourage self-help, Encourage professional help if needed)
-Guidance on setting up the peer support service
*Delivery of training*
MHFA trainers during working day (cover provided for teaching staff)
*Support in developing peer support service*
study team provide written guidance on setting up peer support service and posters to advertise the service

**ii) 1 day MHFA training for schools and colleges**
*Who*
8% teachers in pastoral roles (e.g. tutors) who have not been nominated to be peer supporters
*Content of training*
-Signs and symptoms of common mental disorders
-Case study approach to providing support to young people
-Supporting young people using ALGEE steps
*Delivery of training*
MHFA trainers during In-Service training time

**iii) 1 hour mental health awareness session**
*Who*
All teachers asked to attend, open to all staff
*Content of training*
-Facts about mental health and wellbeing in teenagers and teachers
-The WISE (Wellbeing in Secondary Education) intervention
-What is mental health, stress bucket, five ways to wellbeing, steps to follow to support others
-Local sources of support
*Delivery of training*
MHFA trainers during standard meeting or training time

**Fig 2. Intervention delivery.** ALGEE, Assess for risk, Listen nonjudgmentally, Give information, Encourage self-help, Encourage professional help if needed; MHFA, Mental Health First Aid; WISE, Well-being in Secondary Education.

were set up. Further details of the intervention are shown in Fig 2 and are reported in the protocol [20].

Schools allocated to the comparison group continued with usual practice in terms of teacher support and training. In England and Wales, there is no standard training package for teachers regarding mental health support, although schools are free to choose to pay for any such training that is available. Similarly, there is no standard mental health support available

to staff, although schools are free to buy such a service in, for example, there are several Employee Assistance Programmes (EAPs) available in the UK for workplaces.

## Outcomes

**Primary outcome.**   The primary outcome was teacher well-being, measured using the 14-item Warwick-Edinburgh Mental Well-being Scale (WEMWBS) [25]. The WEMWBS incorporates elements of both subjective and psychological well-being, represented in 14 statements such as "I've been feeling cheerful" and "I've been feeling interested in other people." For each statement, respondents select one out of five options: none of the time/rarely/some of the time/often/all of the time. Each response is scored from 1 (none of the time) to 5 (all of the time), and the total is summed to make up a well-being score of between 14 and 70, where higher scores represent better well-being. The scale has been validated in the general UK adult population, with a Cronbach's alpha score of 0.91, and test–retest reliability at 1 week of 0.83 [21].

**Secondary outcomes.**   Teacher depression was measured using the 8-item Patient Health Questionnaire (PHQ-8) [26]. Items are summed to give a score between 0 and 24; a cut-point of 10 or more indicates moderate to severe depression.

Teacher absence was measured at the individual level by self-report ("number of days missed from school because of health problems in the past four working weeks") and at the school level by data provided by schools (total number of days of absence over the previous year).

Teacher presenteeism was measured by self-report, using the relevant item from the Work Productivity and Activity Impairment (WPAI) questionnaire [27], adapted to fit teachers' work schedule. Teachers were asked "during the last 4 working weeks, how much did health problems affect your productivity while you were working?" and selected a number between 0 (no effect) to 10 (completely prevented me from working).

Teacher retirements and leaving for other reasons were collected at the school level as total number for the previous year.

Student well-being was measured using the 14-item WEMWBS, which has been validated among teenagers from 13 years [28].

Student psychological distress was measured using the Strengths and Difficulties Questionnaire (SDQ), short form version without the impact assessment [29]. A total difficulties score was calculated, which can range from 0 to 40, with a higher score indicating greater difficulties.

Student attendance was collected at the school level from routine data sources (www.compare-school-performance.service.gov.uk for England and www.mylocalschool.wales.gov.uk for Wales).

Student attainment at end of year 11 was collected at the school level from the same routine data sources. Exams at the end of statutory school age (key stage 4) are graded and reported differently for the 2 countries. For English schools, we reported % achieving grade 5 or above in both English and Maths, and for Welsh schools, we reported % achieving grade C or above in English/Welsh, Maths, and Science.

## Process outcomes

Process outcomes were collected in the same teacher and student surveys as the outcome measures. The teacher survey contained 5 questions, and the student survey contained 3 items, all of which were separate items as opposed to items on one scale.

Teacher perceived stress at work: We included the following question in the teacher survey, "In general, how stressful do you find your job?" [response options: not at all stressful, mildly stressful, moderately stressful, very stressful, extremely stressful].

Teacher perceived support and relationships: We included a question in the teacher survey, "Rate how much you agree with the following statements": this school cares about staff well-being/this school cares about student well-being/staff generally have good relationships with each other in this school/teachers and students generally have good relationships in this school [response options: strongly agree, agree, disagree, strongly disagree].

Student perceived support and relationships: We included a question in the student survey, "Rate how much you agree with the following statements": this school cares about student well-being/teachers and students generally have good relationships in this school/my teachers are there for me when I need them [response options: strongly agree, agree, disagree, strongly disagree].

## Economic evaluation

The economic analysis took a public sector perspective, calculating the financial and opportunity costs to schools of participating in the WISE intervention. A proforma was completed after each training session documenting the resources used for trainer fees, HSCs' time (based on salary), trainer and HSC travel expenses, MHFA handbook fees, and the time of the teachers attending training (based on salary).

## Statistical analysis

Full details of the statistical analysis plan are published on the study website (https://www.bristol.ac.uk/population-health-sciences/projects/wise/).

The study was powered to detect a mean change of 3 points on the teacher WEMWBS score, chosen as the minimum meaningful change [30]. We assumed a mean of 50 teacher responses per school (with a coefficient of variation of sample size of 0.5)—which reflects the response rate of approximately 75% of teachers achieved in the pilot study [13]—and an SD for WEMWBS of 8.4 and an intracluster coefficient (ICC) of 0.01 (based on the pilot data [13]). A sample size of 24 schools (12 intervention and 12 control) would achieve 83% power for an ICC of 0.05. Sample sizes were calculated using the Stata clustersampsi command.

Analyses were carried out under the intention to treat principle. For the primary analysis and secondary teacher individual-level outcomes, mixed effects regression was used to compare outcome by arm over the course of follow-up (T0, T1, and T2), adjusted for stratification variables, sex, years of experience, and time. This included a random effect for individual participants, and another for school. Using a maximum likelihood estimator this analysis was robust to data missing at random (MAR) and allowed inclusion of every teacher who had at least one measure of the outcome. The primary analysis was repeated with a treatment by time interaction term added to the model to enable estimation of effect at T1 and T2 separately.

For all student individual-level outcomes and all school-level outcomes, the outcome at T2 was regressed on treatment arm and T0 of the outcome, accounting for clustering by school (using a random effect). All secondary outcome models were adjusted for stratification variables, and the individual student-level models were additionally adjusted for sex and ethnicity.

The impact of missingness and potential departure from MAR (i.e., missing not at random (MNAR)), on teacher WEMWBS and PHQ-8, and student WEMWBS and SDQ outcomes was examined by repeating analyses on multiply imputed data, which incorporated scaling parameters allowed to differ between arms [31].

Complier Average Causal Effect (CACE), using Instrumental Variable analysis [32], was used to investigate efficacy of the intervention (based on completion of MHFA training) for comparison with the primary analysis estimate of effectiveness. Adjustment was made for both school-level and individual-level covariates, as in the primary analysis.

To explore whether the hypothesised mechanisms of change—perceived support to staff and students and quality of staff–student relationships—differed by intervention arm at follow-up, all the process evaluation single items were recoded into binary variables (agree/disagree). Logistic regression models (one per item) were used to compare outcomes for these variables between arms at T2, adjusted for T0 scores, school-level FSM, and geographical area.

Analyses were conducted using Stata (version 14, College Station, TX: StataCorp LP). Findings are reported in line with CONSORT guidelines extension to cluster randomised trials.

We made the following changes to the published protocol [20]: Some levels within the FSM strata were merged during recruitment due to the limited number of schools available (specifically medium and high were merged for the English sampling frame, and medium and low were merged for the Welsh sampling frame); we recruited one extra school in case of drop out, and outcome measures at 3 schools were collected in teachers' own time rather than during meeting time. All changes were approved by the Trial Steering Committee.

## Results

All 25 schools remained in the trial until the end and participated in all 3 data collections. Participant flow through the trial is shown in the CONSORT chart (Fig 3). Some teachers were not eligible for data collection at each time point, for example, if they left the school, joined the school midway through the study, or were on maternity leave. Teacher response rates were above 75% at each time point but were slightly lower in intervention schools at T1 and T2 compared to controls. The total number of teachers included in the primary analysis—that is those who had outcome data at one or more time point—was 1,722.

Schools were well balanced across study arms for most teacher, student, and school-level characteristics at baseline (Table 1). Mean total difficulties score was slightly higher among control school students. Intervention schools had a considerably higher median number of teachers leaving for nonretirement reasons and were more likely to have national average or above in student attainment.

The ICC for the primary outcome teacher well-being at baseline was 0.021 (95% CI 0.007, 0.062).

Table 2 shows the results for all primary and secondary outcomes. There was no evidence of a difference between arms in mean teacher mental well-being over the course of follow-up: unadjusted difference in means −0.48, 95% CI −1.63, 0.67, $p$-value 0.412 and adjusted difference in means −0.90, 95% CI −2.07, 0.27, $p$-value 0.130. There was also no evidence that treatment effect varied over time ($p$-value for interaction term 0.654). Well-being appeared to improve in both arms over time, with a larger improvement in the control arm initially, as shown in Fig 4.

There was no evidence of a difference between intervention and control groups in teacher reported depression or presenteeism. There was weak evidence of a difference between groups in self-reported teacher absenteeism over the course of follow-up; in the adjusted model, the number of days of absence was 4% higher in those teachers in the intervention group compared to those in the control (unadjusted ratio of geometric means 1.03, 95% CI 0.98, 1.07, $p$-value 0.308 and adjusted ratio of geometric means 1.04, 95% CI 1.00, 1.09, $p$-value 0.042). There was no evidence of a difference in student mental health outcomes (WEMWBS or SDQ) between intervention and control groups at T2.

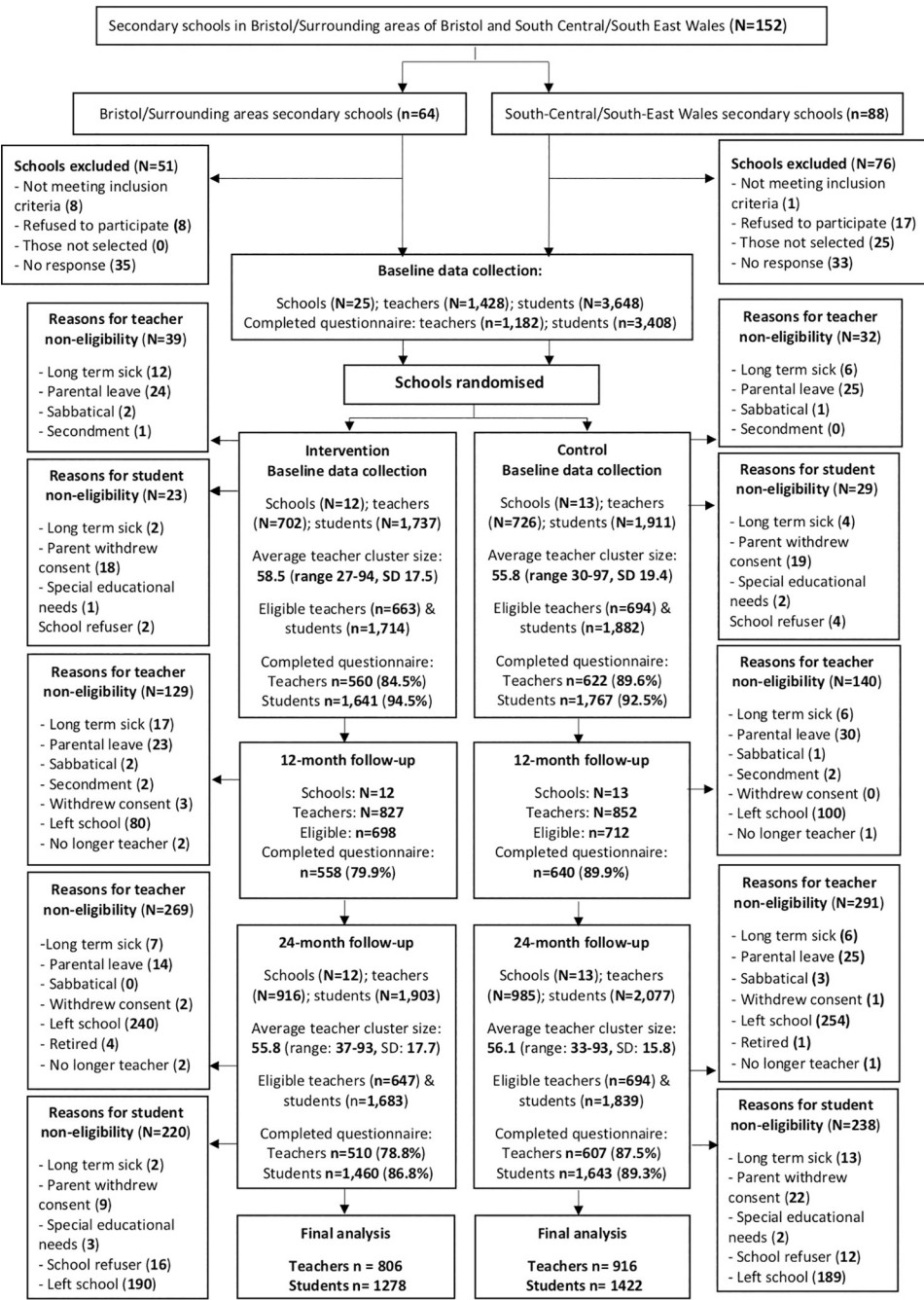

**Fig 3. Participant flow through the study.**

There was also no evidence of a difference between intervention and control groups in any of the school recorded outcomes (teacher absenteeism, teacher retirement, teachers leaving for other reasons, student attainment, and student attendance). There were more teachers retiring in the intervention schools, although this may be explained by participants being older (56% of teachers in the intervention arm had worked in a school for more than 10 years compared to 52% in the control arm at T0).

**Table 1. Comparison of baseline characteristics between arms.**

| | Control | Intervention |
|---|---|---|
| *Teacher variables* | | |
| Male (%) | 221 (36%) | 205 (37%) |
| WEMWBS mean (SD) score | 46.8 (8.2) | 46.8 (8.6) |
| PHQ-8 median (IQR) score | 5 (3, 9) | 5 (3, 9) |
| Presenteeism median (IQR) score | 1 (0, 3) | 1 (0, 3) |
| Median days absent (IQR) past 4 weeks | 0 (0, 0) | 0 (0, 0) |
| Ever absent in the previous 4 weeks (%) | 101 (16%) | 83 (15%) |
| Previous mental health problem (%) | 223 (37%) | 189 (34%) |
| Previous MH training (%) | 11 (2%) | 7 (1%) |
| Satisfied or very satisfied with job (%) | 348 (55%) | 316 (57%) |
| Job moderately to extremely stressful (%) | 522 (85%) | 453 (81%) |
| Length of time working in a school (%) | | |
| • <1 year | 41 (7%) | 24 (4%) |
| • 1–5 years | 121 (20%) | 98 (17%) |
| • 6–10 years | 133 (22%) | 122 (22%) |
| • >10 years | 319 (52%) | 315 (56%) |
| White (%) | 591 (97%) | 539 (97%) |
| Provided support to a colleague more than once a month in the last year (%) | 288 (46%) | 269 (48%) |
| Good relationship between students and teachers in the school (%) | 602 (98%) | 538 (96%) |
| Good relationship between staff (%) | 570 (92%) | 513 (93%) |
| *Student variables* | | |
| Male (%) | 918 (47%) | 884 (49%) |
| SDQ mean (SD) score | 12·7 (6·1) | 12·0 (6·0) |
| Student WEMWBS mean (SD) score | 47·0 (9·4) | 47·5 (9·0) |
| Went to a teacher for help with social/personal problem once a term or more in the past year (%) | 211 (12%) | 180 (11%) |
| Wanted to ask for help from a teacher at least once a term in the past year but felt unable to (%) | 195 (14%) | 170 (14%) |
| Good relationship between students and teachers in the school (%) | 1,207 (70%) | 1,209 (75%) |
| Ethnicity (%): | | |
| • White | 1,668 (85%) | 1,448 (81%) |
| • Mixed | 147 (8%) | 147 (8%) |
| • Asian or Asian British | 69 (4%) | 91 (5%) |
| • Black or black British | 45 (2%) | 78 (4%) |
| • Chinese | 24 (1%) | 23 (1%) |
| *School-level variables* | | |
| Median (IQR) teacher student ratio | 0.33 (0.31, 0.36) | 0.35 (0.31, 0.39) |
| Median % (IQR) of teachers retired over past year | 1.7 (0.7, 3.2) | 0 (0, 1.8) |
| Median % (IQR) of teachers left for other reasons over past year | 14.6 (7.3, 21.4) | 21.6 (11.7, 25.0) |
| Mean number (range) of teachers in whole school | 60.1 (33, 95) | 60.0 (27, 101) |
| FSM tertile (%) | | |
| • Low | 3 (23%) | 6 (50%) |
| • Middle | 6 (46%) | 2 (17%) |
| • High | 4 (31%) | 4 (33%) |
| Mean number (range) of students in whole school | 834.3 (484, 1,203) | 907.2 (389, 1,584) |
| Median % (IQR) teacher absence over past year | 6.1 (3.7, 12.8) | 4.0 (3.5, 10.1) |
| Median % student attendance | 93.6 (93.4, 94.8) | 93.7 (93.2, 94.8) |

*(Continued)*

**Table 1.** (Continued)

|  | Control | Intervention |
|---|---|---|
| Number (%) of schools > = average student attainment | 4 (30%) | 5 (42%) |

FSM, free school meals; IQR, interquartile range; MH, mental health; PHQ-8, Patient Health Questionnaire 8-item; SD, standard deviation; SDQ, Strengths and Difficulties Questionnaire; WEMWBS, Warwick Edinburgh Mental Well-being Scale.

Under an MAR assumption, there was no difference in conclusion drawn from the analysis of the imputed data compared to the primary analysis (Tables A-D in S1 Text).

There was no evidence of a treatment effect for the primary outcome WEMWBS in those who attended MHFA training, although there was a weak tendency in the direction favouring participants in the control schools in the adjusted analysis (unadjusted mean difference −2.51, 95% CI −7.51, 2.49, $p = 0.326$, adjusted mean difference −2.26, 95% CI −4.93, 0.40, $p = 0.096$).

Table 3 shows the results for all mechanisms of change outcomes. There was no evidence of a difference between intervention and control groups at T2 with respect to teacher self-reported stress at work, or teachers' perceptions of school attitude towards student well-being, and quality of relationships between staff or between teachers and students. In the unadjusted model, teachers in the control group were more likely to agree that school cares about staff well-being (OR = 0.48, 95% CI 0.28, 0.82, $p = 0.01$), but this difference was attenuated once adjustment had been made for T0 scores and stratifying variables (adjusted OR = 0.58, 95% CI 0.29, 1.19, $p = 0.14$). There was also no difference by study arm at T2 in student perceptions of whether their school cares about student well-being or teachers being there for them when needed. There was a difference in favour of the control arm in students who agreed that their school had good teacher-student relationships in both the unadjusted and adjusted models (unadjusted OR = 0.82, 95% CI 0.70, 0.95, $p = 0.01$ and adjusted OR = 0.80, 95% CI 0.66, 0.96, $p = 0.02$).

The average cost of the intervention was £9,103 per intervention school. The cost was slightly lower in English schools (£8,263) than in Welsh schools (£9,943), primarily due to the upfront cost of training the HSCs in Wales. Staff salaries and costs for supply teachers (£5,566) accounted for the majority (61%) of the total cost of the intervention. There was no evidence that this additional cost was justified by improvement in teacher well-being, depressive symptom scores, presenteeism or absenteeism, or by any of the student reported outcomes.

## Discussion

We found no evidence that the WISE intervention had an effect on the primary outcome teacher well-being; in both arms, there was an increase in well-being over time, with this increase being larger in control schools in the first year of the intervention. We also found no intervention effects for teacher depression or presenteeism, or student well-being, psychological difficulty, attendance, or attainment. There was one small effect in favour of the control arm for teacher absence in the fully adjusted models. However, this was not a large effect and may have been a chance finding due to the number of analyses undertaken. Although the total mean cost of the intervention per school was relatively small, we found insufficient evidence that it was justified by any improvements in staff or student outcomes at the individual or school level.

Our process evaluation findings showed no difference between groups at the study end regarding teacher stress at work, how far teachers felt schools cared about well-being, and teachers' perceptions of the quality of relationships. Among students, there was a general

**Table 2. Results for primary and secondary outcomes[a].**

| Primary outcome | T0 | | T1 | | T2 | | Unadjusted difference in means (95% CI) | p-value | Adjusted[b] difference in means (95% CI)[c] | p-value |
|---|---|---|---|---|---|---|---|---|---|---|
| | Control | Intervention | Control | Intervention | Control | Intervention | | | | |
| Teacher WEMWBS score mean (SD) (N = 1,722)[d] | 46.8 (8.2) | 46.8 (8.6) | 48.1 (8.7) | 47.4 (9.4) | 48.4 (8.4) | 47.5 (8.6) | −0.48 (−1.63, 0.67) | 0.412 | −0.90 (−2.07, 0.27) | 0.130 |
| **Secondary teacher outcomes** | | | | | | | **Ratio of geometric means intervention/ control (95% CI)[e]** | **p-value** | **Ratio of geometric means intervention/ control (95% CI)[b,e]** | **p-value** |
| PHQ-8 score median (IQR) (N = 1,719)[d] | 5 (3, 9) | 5 (3, 9) | 5 (2, 9) | 5 (2, 9) | 5 (2, 8) | 5 (2, 9) | 0.97 (0.88, 1.07) | 0.577 | 1.00 (0.92, 1.10) | 0.964 |
| Number of days absent in past 4 weeks median (IQR) (N = 1,717)[d] | 0 (0, 0) | 0 (0, 0) | 0 (0, 0) | 0 (0, 0) | 0 (0, 0) | 0 (0, 0) | 1.03 (0.98, 1.07) | 0.308 | 1.04 (1.00, 1.09) | 0.042 |
| | | | | | | | **Difference in means (95% CI)** | **p-value** | **Adjusted[b] difference in means (95% CI)[c]** | **p-value** |
| Presenteeism score median (IQR) (N = 1,539)[d] | 1 (0, 3) | 1 (0, 3) | 1 (0, 3) | 1 (0, 3) | 1 (0, 3) | 1 (0, 3) | 0.03 (−0.26, 0.32) | 0.862 | 0.12 (−0.13, 0.37) | 0.361 |
| **Secondary student outcomes** | | | | | | | **Difference in means (95% CI)** | **p-value** | **Adjusted[f] difference in means (95% CI)[c]** | **p-value** |
| WEMWBS score mean (SD) (N = 2,700)[d] | 47.0 (9.4) | 47.5 (9.0) | | | 44.9 (9.7) | 45.0 (10.0) | 0.29 (−0.93, 1.50) | 0.642 | −0.35 (−1.35, 0.66) | 0.500 |
| SDQ score mean (SD) (N = 2,702)[d] | 12.7 (6.1) | 12.0 (6.0) | | | 13.5 (6.0) | 13.3 (5.9) | −0.27 (−1.25, 0.70) | 0.583 | 0.27 (−0.18, 0.73) | 0.241 |
| **Secondary school-level outcomes** | | | | | | | **Difference in means (95% CI)** | **p-value** | **Adjusted[g] difference in means (95% CI)[c]** | **p-value** |
| Median % (IQR) teacher absence in past year (N = 13)[d] | 6.1 (3.7, 12.8) | 4.0 (3.5, 10.1) | | | 4.8 (3.4, 7.6) | 7.9 (2.2, 9.9) | 0.41 (−5.31, 6.13) | 0.878 | 3.69 (−1.72, 9.11) | 0.151 |
| Median % (IQR) teachers retired in past year (N = 8)[d] | 1.7 (0.7, 3.2) | 0 (0.0, 1.8) | | | 0.7 (0.0, 2.2) | 8.4 (3.6, 13.2) | 7.43 (−2.15, 17.01) | 0.107 | 10.78 (−2.17, 23.73) | 0.070 |
| Median % (IQR) teachers left for other reasons in past year (N = 8)[d] | 14.6 (7.3, 21.4) | 21.6 (11.7, 25.0) | | | 14.0 (10.4, 20.0) | 43.9 (3.6, 84.2) | 28.27 (−52.75, 109.29) | 0.426 | 47.36 (−148.39, 243.11) | 0.407 |
| Median % (IQR) student attendance in past year (N = 25)[d] | 93.6 (93.4, 94.8) | 93.7 (93.2, 94.8) | | | 93.5 (92.8, 94.2) | 93.2 (92.6, 94.5) | 0.04 (−1.06, 1.14) | 0.942 | 0.17 (−0.60, 0.95) | 0.645 |
| | | | | | | | **OR (95% CI)** | **p-value** | **Adjusted[g] OR (95% CI)[c]** | **p-value** |
| Number (%) of schools with student attainment > = average (N = 9)[d,h] | 4 (30%) | 5 (42%) | | | 3 (23%) | 4 (33%) | 0.60 (0.10, 3.49) | 0.570 | 1.12 (0.03, 38.72) | 0.950 |

[a]Where distribution was skewed, scores at each time point are reported as median and IQR.

[b]Adjusted for region, FSM, gender, years of experience, and time.

[c]Control group as reference.

[d]Number in fully adjusted analysis.

[e]Where model assumptions were violated, secondary outcomes were log transformed and regression coefficients are reported as a ratio of geometric means.

[f]Adjusted for baseline score, region, FSM, gender, and ethnicity.

[g]Adjusted for baseline score, region, and FSM.

[h]Data were available for all 25 schools, but some observations had to be omitted from the analysis due to issues of perfect prediction and collinearity.

CI, confidence interval; FSM, free school meal; IQR, interquartile range; N, number; PHQ-8, Patient Health Questionnaire 8-item; OR, odds ratio; SD, standard deviation; SDQ, Strengths and Difficulties Questionnaire; T0, baseline; T1, time 1 follow-up; T2, time 2 follow-up; WEMWBS, Warwick Edinburgh Mental Well-being Scale.

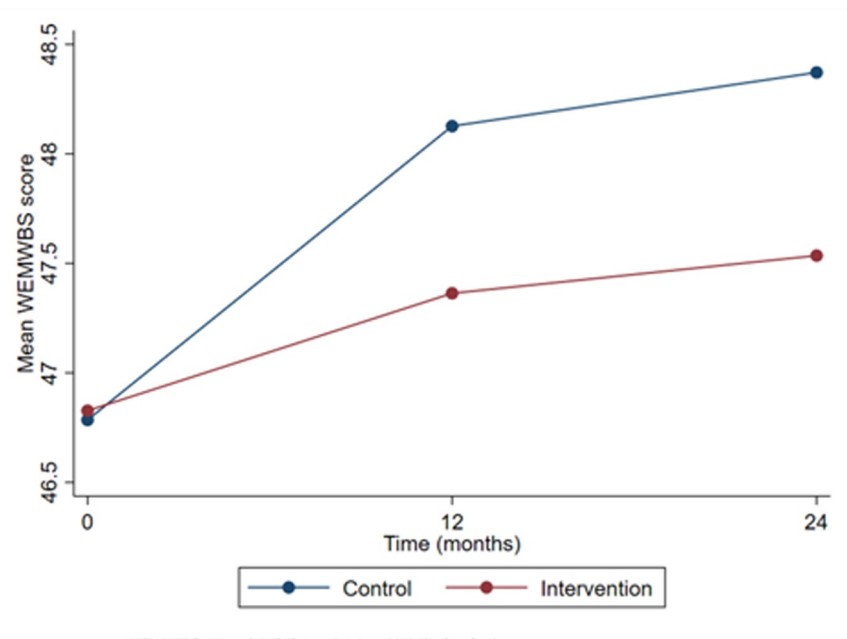

WEMWBS; Warwick Edinburgh Mental Wellbeing Scale

**Fig 4. Teacher well-being over time.** WEMWBS, Warwick-Edinburgh Mental Well-being Scale.

**Table 3. Results for mechanisms of change variables.**

| Mechanism of change variable | T0 | | T2 | | Undjusted[a] OR (95% CI)[b] | p-value | Adjusted OR (95% CI) | p-value |
|---|---|---|---|---|---|---|---|---|
| | Control N (%) | Intervention N (%) | Control N (%) | Intervention N (%) | | | | |
| *Teacher survey (N in fully adjusted analyses)* | | | | | | | | |
| Agree job is stressful (N = 694) | 522 (83.9) | 451 (81.3) | 458 (75.6) | 391 (76.8) | 0.91 (0.62, 1.33) | 0.62 | 1.02 (0.63, 1.67) | 0.93 |
| Agree school cares about staff well-being (N = 675) | 376 (61.8) | 297 (53.7) | 433 (71.8) | 278 (55.5) | 0.48 (0.28, 0.82) | 0.01 | 0.58 (0.29, 1.19) | 0.14 |
| Agree school cares about student well-being (N = 689) | 587 (95.1) | 522 (93.9) | 350 (93.8) | 294 (91.3) | 0.34 (0.11, 1.03) | 0.06 | 0.48 (0.14, 1.61) | 0.23 |
| Agree staff generally have good relationships (N = 688) | 570 (92.2) | 513 (92.3) | 569 (94.7) | 477 (93.7) | 0.83 (0.31, 2.21) | 0.71 | 0.49 (0.22, 1.10) | 0.08 |
| Agree teacher-students generally have good relationships (N = 657) | 602 (97.4) | 538 (96.6) | 359 (96.0) | 312 (96.9) | 0.73 (0.25, 2.17) | 0.57 | 01.17 (0.35, 3.92) | 0.80 |
| *Student survey (N in fully adjusted analyses)* | | | | | | | | |
| Agree school cares about student well-being (N = 2,660) | 1,213 (85.7) | 1,119 (87.4) | 1,097 (76.9) | 959 (75.0) | 0.87 (0.74, 1.03) | 0.11 | 0.88 (0.72, 1.08) | 0.22 |
| Agree teacher–students generally have good relationships (N = 2,636) | 1,007 (71.2) | 977 (76.3) | 1,023 (72.3) | 854 (67.5) | 0.82 (0.70, 0.95) | 0.01 | 0.80 (0.66, 0.96) | 0.02 |
| Agree teachers are there for them when they need them (N = 2,575) | 1,043 (74.7) | 994 (78.9) | 994 (71.5) | 899 (71.3) | 0.99 (0.84, 1.17) | 0.92 | 1.02(0.84, 1.25) | 0.83 |

[a]Adjusted for T0 scores, region, and school FSM.

[b]Control group as reference.

CI, confidence interval; FSM, free school meal; N, number; OR, odds ratio; T0, baseline; T2, time 2 follow-up.

pattern across both groups of a decline in positive perceptions regarding how far school cares about well-being and whether teachers are there for them when needed. Intervention students were also less likely to agree that teachers and students generally have positive relationships by time two, whereas control students' views remained constant across the time points. In other words, the intervention did not appear to have had an impact on the mechanisms of change outlined in our logic model, which is likely to at least partially explain the lack of effectiveness of the intervention.

Strengths of the study include the high response rates from both teachers and students, and retention of all recruited schools, which ensured sufficient statistical power to identify important differences in mental health outcomes between study arms. High response rates and the use of multiple imputation in our analysis helped limit the risk of reporting bias due to missing outcome data. The collection of baseline data before randomisation further helped to mitigate risk of bias. The stratified approach to recruitment helped ensure balance across study arms at baseline, with the exception of student attainment and teachers leaving for reasons other than retirement. This stratification process also ensured that the study sample spanned the full range of socioeconomic catchment areas of schools in England and Wales (as measured by FSM eligibility), and results were similar in both the English and the Welsh schools, suggesting that findings are likely to be generalisable across all mainstream schools in these 2 countries.

The lack of blinding of participants was a limitation, as we cannot be sure that knowing they were in an intervention school did not affect teachers' responses to the self-report surveys. For example, if participating in the intervention led to raised expectations regarding changes in school, and if these expectations were not perceived to be met, then teachers in intervention schools may have given more negative responses about their school environment or their own mental health and well-being at follow-up. A second problem with using self-report measures of well-being and mental health is the risk that gaining knowledge about mental health through the intervention itself may affect how individuals then report their own mental health [33]. A third limitation for school-level outcomes was that we did not always have data from all schools, leading to a lack of precision in some of the secondary outcomes, and potential for bias if these were not MAR. Finally, although we planned to work with school link workers based within CAMHS in England and with the HSCs in Wales to provide ongoing support to the peer supporters, this did not happen in practice. In England, this was because the school link worker role was not filled for many of our study schools, and in Wales, this was because the HSCs who had the capacity to take on the trainer roles were not those who were supporting the study schools. This limitation is unlikely to explain the study's main finding that overall teacher well-being did not improve, but it may have contributed to the peer support services losing some sense of momentum and being fully embedded in school life, as discussed below.

There are a number of possible explanations for the null finding reported here. First, the intervention may not have been sufficiently differentiated from usual practice, particularly given the context of the UK government's focus on schools as a setting for improving mental health support [34]. We did see improvements in teacher well-being in both arms, as shown in Fig 4, which may be due to an effect of national policy, although the lack of prescriptive guidance on *how* schools should support teacher mental health training and a lack of focus on teacher well-being perhaps makes this unlikely. A second explanation is that a MHFA training and peer support intervention is not an effective way in which to address the mental health training and support needs of teachers. Findings from our pilot study indicated that the MHFA training was effective at improving knowledge, skills, and confidence in supporting mental health and that the peer support service increased the opportunity for staff to gain support from colleagues [13]. However, evidence from the process findings reported here indicate that this increase in knowledge, skills, confidence, and available support did not translate into

change at the population and whole-school level regarding improved experience of support for mental health, improved quality of relationships, or reduced stress in the workplace. One reason for this may have been a problem with the implementation of the intervention. Previously published evidence from this part of our process evaluation reported implementation to be good overall, but it also revealed some difficulties both in delivering the full length, uninterrupted training to enough staff in all schools, and in embedding the peer support service into school life [24]. This led to limitations in dose and reach which may have meant that the intervention as it was delivered in practice was not sufficient to create whole-school, sustained changes, although it may have had positive effects for certain individuals.

Reflecting further on these implementation findings, it is possible that the context in which the intervention was delivered contained barriers both to its successful implementation and to the intervention having a large enough impact. It is increasingly acknowledged that context must be taken into account when evaluating real world public health interventions, due to its key role in determining the success of an intervention being implemented and having an impact [35]. In the case of schools, various aspects of the context have been found to be associated with both implementation of mental health interventions [36] and with student and teacher mental health outcomes [2,37]. We attempted to address such contextual factors in the present intervention through strengthening supportive relationships and giving mental health a higher profile, but our process outcome findings indicate that it was not successful in this regard. We reported previously that the challenges encountered in implementation were due at least in part to support from senior leadership waning over time [24]. Visible and meaningful support from senior leaders is likely to be an important factor in the creation of a school culture that enables mental health interventions to thrive and have an impact. Our sample of teachers had poor well-being compared to the general UK working population [5], and, elsewhere, teachers have noted the challenging environment in which they work [3,38,39], describing a culture of not feeling able to ask for help or disclose problems [1,3]. If those in the intervention schools had the promise of improved support, only to find that not enough changed in terms of the challenging culture, it is perhaps not surprising that there was no sizeable improvement in well-being.

In conclusion, poor teacher mental well-being and the knock-on effects for student learning and mental health remain important public health issues, particularly in the aftermath of the COVID-19 pandemic and related disruptions to school life, which have created additional stressors for teachers [40] and concerns for children and young people's mental health and well-being [41]. Existing evidence indicates that peer support is important for well-being in the workplace [42–44], including for teachers [12, 45], but rigorous evaluation of workplace peer support interventions is lacking. Further, a need for improving teachers' skills in supporting student mental health has been identified [3,34], but previous studies have focused on this without also explicitly including support for teacher mental health [22]. To our knowledge, we conducted the first ever cluster RCT of an intervention to improve teacher and student mental health, comprising a staff peer support service and MHFA training. The intervention was not effective compared to usual practice, possibly due to the challenges of introducing an intervention that has a large and sustainable impact at the whole-school level. More research is needed to identify effective approaches to improve teacher mental health. This should include interventions that focus on addressing the challenging context and culture of schools, which can often be detrimental to mental health and well-being. Policy changes to reduce the structural determinants of poor teacher mental health—including high workload and performance management based on academic outcomes—and assessment of mental health support as part of government inspections of schools to ensure that it is prioritised would help create a wider context in which mental health interventions could prosper.

## Supporting information

**S1 Text. Supporting information: results from imputed texts.** Table A. Results from imputed models for teacher WEMWBS outcome. Table B. Results from imputed models for teacher PHQ-8 outcome. Table C. Results from imputed models for student WEMWBS outcome. Table D. Results from imputed models for student SDQ outcome.
(DOCX)

**S1 Table. CONSORT Checklist.**
(DOCX)

## Acknowledgments

The work was undertaken with the support of The Centre for the Development and Evaluation of Complex Interventions for Public Health Improvement (DECIPHer), a UK Clinical Research Collaboration (UKCRC) PHR Centre of Excellence. This study was designed and delivered in collaboration with the Bristol Randomised Trials Collaboration (BRTC), a UKCRC Registered Clinical Trials Unit in receipt of NIHR Clinical Trials Unit support funding, and we gratefully acknowledge support from Jodi Taylor in providing ongoing trials advice.

We also extend our thanks to the Schools Health Research Network in Wales and local public health partners in England for their support in recruiting schools. We gratefully acknowledge the invaluable contribution of Hannah Baber (Trial Manager from 2018). We thank Aideen Ahern for contributing to the design and conduct of the economic analysis, Dr Mari-Rose Kennedy for support with data analysis, and Professor Kate Tilling for early advice regarding study design and statistical analysis plan. Finally, our thanks go to all the staff and student participants of the study, and the MHFA trainers and Healthy Schools Coordinators for their contributions to intervention delivery.

## Author Contributions

**Conceptualization:** Judi Kidger, William Hollingworth, Rhiannon Evans, Ricardo Araya, Rona Campbell, Tamsin Ford, David Gunnell, Simon Murphy.

**Data curation:** Harriet Fisher.

**Formal analysis:** Judi Kidger, Nicholas Turner, William Hollingworth, Sarah Bell, Lauren Copeland, Harriet Fisher, Sarah Harding, Richard Morris.

**Funding acquisition:** Judi Kidger, William Hollingworth, Rhiannon Evans, Ricardo Araya, Rona Campbell, Tamsin Ford, David Gunnell, Simon Murphy.

**Investigation:** Judi Kidger, Rhiannon Evans, Sarah Bell, Rowan Brockman, Sarah Harding, Jillian Powell.

**Methodology:** Judi Kidger, Nicholas Turner, William Hollingworth, Rhiannon Evans, Sarah Bell, Ricardo Araya, Rona Campbell, Tamsin Ford, David Gunnell, Simon Murphy, Richard Morris.

**Writing – original draft:** Judi Kidger, Nicholas Turner, Richard Morris.

**Writing – review & editing:** William Hollingworth, Rhiannon Evans, Sarah Bell, Rowan Brockman, Lauren Copeland, Harriet Fisher, Sarah Harding, Jillian Powell, Ricardo Araya, Rona Campbell, Tamsin Ford, David Gunnell, Simon Murphy.

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
