## [Editor Report · Decision Letter 0]

4 May 2021

Dear Dr Kidger, 

Thank you for submitting your manuscript entitled "An intervention to improve high school teacher wellbeing support and training (the WISE study): a cluster randomised controlled trial" for consideration by PLOS Medicine.

Your manuscript has now been evaluated by the PLOS Medicine editorial staff and I am writing to let you know that we would like to send your submission out for external peer review.

Please re-submit your manuscript within two working days, i.e. by May 06 2021 11:59PM.

Kind regards,

Beryne Odeny

Associate Editor

PLOS Medicine

---

## [Decision Letter · Decision Letter 1]

22 Jun 2021

Dear Dr. Kidger,

Thank you very much for submitting your manuscript "An intervention to improve high school teacher wellbeing support and training (the WISE study): a cluster randomised controlled trial" (PMEDICINE-D-21-02018R1) for consideration at PLOS Medicine. 

[LINK]

In light of these reviews, I am afraid that we will not be able to accept the manuscript for publication in the journal in its current form, but we would like to consider a revised version that addresses the reviewers' and editors' comments. Obviously we cannot make any decision about publication until we have seen the revised manuscript and your response, and we plan to seek re-review by one or more of the reviewers. 

We expect to receive your revised manuscript by Jul 13 2021 11:59PM. Please email us (plosmedicine@plos.org) if you have any questions or concerns.

We look forward to receiving your revised manuscript. 

Sincerely,

Beryne Odeny, 

PLOS Medicine

plosmedicine.org

1) Please include the study setting in the title before the colon. 

2) The Data Availability Statement (DAS) requires revision: If the data are not freely available, please include an appropriate email address for inquiries (this cannot be a study author). 

3) Author summary - At this stage, we ask that you reformat your non-technical Author Summary. The Author Summary should immediately follow the Abstract in your revised manuscript. This text is subject to editorial change and should be distinct from the scientific abstract. The summary should be accessible to a wide audience that includes both scientists and non-scientists. Please see our author guidelines for more information: https://journals.plos.org/plosmedicine/s/revising-your-manuscript#loc-author-summary.

4) Abstract:

a) Please report your abstract according to CONSORT for abstracts, following the PLOS Medicine abstract structure (Background, Methods and Findings, Conclusions) http://www.consort-statement.org/extensions?ContentWidgetId=562

b) Please combine the Methods and Findings sections into one section, “Methods and findings”. 

c) Please ensure that all numbers presented in the abstract are present and identical to numbers presented in the main manuscript text.

d) Please quantify the main results (with 95% CIs and p values).

e) Please include the actual amounts or percentages of all relevant outcomes, not just coefficients.

f) Please include a summary of adverse events if these were assessed in the study.

g) In the last sentence of the Abstract Methods and Findings section, please describe the main limitation(s) of the study's methodology.

h) Please mention specific study implications without overreaching what can be concluded from the data; the phrase "In this study, we observed ..." may be useful.

5) At this stage, we ask that you include a short, non-technical Author Summary of your research to make findings accessible to a wide audience that includes both scientists and non-scientists. The Author Summary should immediately follow the Abstract in your revised manuscript. This text is subject to editorial change and should be distinct from the scientific abstract. Please see our author guidelines for more information: https://journals.plos.org/plosmedicine/s/revising-your-manuscript#loc-author-summary.

6) Please replace the term “cost-effectiveness” with “cost evaluation” or similar as no formal cost-effectiveness analysis was done.

7) Please complete the CONSORT checklist and ensure that all components of CONSORT are present in the manuscript, including [how randomization was performed, allocation concealment, blinding of intervention, definition of lost to follow-up, power statement]. When completing the checklist, please use section and paragraph numbers, rather than page numbers. 

8) Please provide 95% CIs and p values for all estimates in the text and tables.

9) For adjusted analyses, please also provide the unadjusted analyses in the text and tables.

10) In the Discussion please talk about what the study adds to existing research and where and why the results may differ from previous research in similar settings; implications and next steps for research, clinical practice, and/or public policy; one-paragraph conclusion.

11) Please use the "Vancouver" style for reference formatting and see our website for other reference guidelines. 

a) Please ensure that journal name abbreviations match those found in the National Center for Biotechnology Information (NCBI) databases, and are appropriately formatted and capitalized. https://journals.plos.org/plosmedicine/s/submission-guidelines#loc-references. Please ensure that weblinks are current and accessible to date.

b) Six names should precede et al, in all your references.

c) Please include access dates for all weblinks and ensure that all weblinks are current and accessible.

12) Please include line numbers in your next draft

Comments from the reviewers:

Reviewer #1: Dear Author

It is a good article , The title is not adjusted with the procedure ( It the title , the students has not entitled but the authors are mentioned about students in the Methodology).

It would rather to write the students in the title? Whether the teacher wellbeing support affects the mental health of students. Please clarify the students role in the title. while the authors assess and studies the teacher wellbeing support on students

Reviewer #2: Statistical review

This paper reports a cluster randomised trial investigating a mental health intervention aimed at teachers in UK secondary schools. 

Generally the trial was very well reported and I have only a few minor comments:

1. Abstract - given that it's a prospective trial, I would recommend p-values are provided in addition to the confidence intervals, especially for the primary endpoint.

2. Page 5 - in the randomisation process, was a block randomisation used within strata? 

3. Page 9 "To explore whether perceived support to staff and students, and quality of staff-student relationships might explain any differences in outcome by arm" I would recommend rewording this slightly, as it currently implies a mediation analysis.

4. Page 9 "some strata were merged during recruitment due to the limited number of schools available" - can more information on this be provided, is this referring to the randomisation? I had thought there were only two strata (location and FSM level)?

5. Page 10 "(data available as supplementary material)" it may be the authors meant this could be added as supplementary material if necessary. I would recommend it is added. If it meant otherwise then I could not see supplementary material with the submission.

6. I would recommend that the estimated ICC is reported, at least for the primary outcome, as it would help future research.

James Wason

Reviewer #3: Reviewer report:

Reviewer (Dr. Allen Nsangi): Overall, I think the results of the trial are impressive and clear. I think the authors do a good job of explaining their findings, possible mitigating factors and a well-thought through conclusion. However, I find some of the information particularly in the methods section to be incomplete. I think some minor revisions to make the methodology more transparent would greatly improve the manuscript.

Abstract:

1) The abstract is clearly written.

Methods:

2. Study design and participants

"Twenty five mainstream secondary schools (students aged 11-18) were recruited in two

geographical areas… (Page 4 of the manuscript)". While I clearly appreciate the need to summarize information which may already be published in study protocols etc, readers should not be expected to consult the protocols to find basic information about the study context/population. I suggest that you specify in about a sentence how the geographical locations were selected and why.

3. Randomisation and masking

"Within each geographical area and stratum, selected schools were randomly allocated to a

study arm…(page 5 of the manuscript)". I would appreciate a bit more detail on how you ensured equal distribution of schools to the two variables (location and income) in the two study arms.

4. Outcomes

I suggest the authors clearly separate the primary outcomes from the secondary outcomes. It's not clear on the onset if all the listed outcomes are considered primary or not according to the way they are laid out in this section here.

Reviewer #4: This an important study that was well designed, the WISE intervention well described and the whole study was very well implemented. The changes made to the published protocol are acknowledged and these changed were approved by he Trial Steering Committee. The manuscript is also very well written. The hypothesis is clear and so is the designed cluster randomized controlled trial, not blinded The reasons why the schools and the study were not blinded are clear in that it was not possible to blind them because of the nature of the knowledge of the intervention and its delivery at cluster level. The statistical analysis is well described as published on the study website. Even though there was no evidence of an effect of the intervention on the primary outcome teacher well being, nor on the secondary outcomes for both teachers and students this was an excellent paper worth publishing. 

Reviewer #5: Thank you for the opportunity to review this paper, which evaluates the effectiveness of teacher wellbeing and mental health intervention, and as a result student mental health in Wales and England. This paper also estimates the associated costs with the intervention implementation in school setting. Among the adolescent health community, there is a need for more intervention studies focusing on the teachers and other community members that may aid in mental health disorders prevention and management among children and young people. 

The Methods and Analysis utilized are appropriate for the aims of the study and clearly explained. In most sections, however, there are not enough details. The conceptual framework presented for the intervention lacks details regarding the complex relationships between various pathways of mechanisms and trial outcomes. In the Discussion, the authors fail to provide thoughtful explanations for the study findings and draw on previously published literature to support or refute the current findings.

Below are a few comments for consideration.

* Abstract: The Methods section of the abstract should provide more information about the cluster trial (e.g., allocation ratio, number of clusters, number of participants, stratification variables, etc.) Secondary outcomes are not clear. Include the p values for the differences between the two arms and the regression models used for the analyses.

* Evidence in the introduction section is a bit outdated and irrelevant. Caldwell and others' systematic review, for example, is focused on universal and targeted interventions for children and adolescents in schools. It might be worthwhile for the authors to review the evidence on interventions for teachers (Yamaguchi et al 2020). Both Bonnell et al and Ford et al focus on whole-school interventions rather than teacher-targeted interventions. They have a few components that focus on teacher training. There is no clear connection between intervention mechanisms and outcomes in the logic model (Fig. 1). Provide more current evidence with links between intervention mechanisms and outcomes. Who were the healthy school coordinators and the CAMHS link workers? What was their role in implementing the intervention? A detailed profile of this human resource's background is needed. What was the intervention of the control arm?

* There is also no information provided on the acceptability and feasibility of the intervention and its adaptation to the school setting. The latter is briefly discussed as one of the limitations of the study. 

* There is a need for supportive literature for the second hypothesis. How improving the mental health and wellbeing of teachers can improve relationships between teachers and students and hence result in better mental health for students? The logical framework fails to recognize the complex relationships between various levels of outcomes. The purpose of clubbing process outcomes with outcomes is not clear. As described in the hypothesis, a positive student-teacher relationship may improve student mental health as a result of better mental health of teachers. 

* More details on the trial design are required. How many schools were randomized to each arm? What was the distribution based on two geographic locations? What was the randomization method? What are 'mainstream' and 'alternative provision' secondary schools? What do low, medium, and high levels of FSM mean?

* Were there any inclusion and exclusion criteria for students?

* How was the opt-out assent procedure employed for teachers and students?

* Were the student assessments also paper-based? Who administered the questionnaires to the teachers and students? Were teachers present when the student assessment carried out in the tutor groups/lesson time? Were the data collectors masked to the school's allocation status? If yes/no, reasons for it.

* Which were the primary outcome(s) and secondary outcomes?

* Which version of the WEMWBS was used? Please provide further details on WEMWBS items, scoring, and validity of WEMWBS in the adult target population.

* Although the authors mentioned that the details on intervention are published elsewhere, some details on the intervention mechanism, broad topics, and strategies of training, intervention active duration, support and supervision in offering peer support service and establishing further services for students, etc. are needed. 

* Clarification on which version of the SDQ was used is required.

* How were the items related to perceived support and relationships were scored? What were the possible minimum and maximum scores?

* Details on the process evaluation are required (e.g., frequency, data collection method, who collected process data, indicators selected, etc.)

* Economic evaluation: it is not appropriate to call the cost-estimation exercise a cost-effectiveness evaluation. 

* Did the authors account for loss to follow-up in the power calculation? Were the assumptions for power calculation rooted in actual data? Provide the citation for ICC value. 

* Statistical analyses: Mention the way findings are reported. 

* Why are the process evaluation results not presented?

* What was the ICC for the primary outcome at baseline?

* What kinds of severe adverse events observed? 

* There is not enough and convincing discussion of no impact of the intervention on outcomes or mechanisms. The process evaluation results are not presented and discussed sufficiently to unpack the findings of the trial. 

* Why do the authors consider the non-blinding of the participants as one of the limitations of the study? How would knowing the intervention allocation status influence the responses of teachers and students?

* The authors have discussed the school context as one of the limitations of the study. However, it should be presented as an explanation for receiving the negative findings among many other potential reasons. 

* In the absence of sufficient information on control arm intervention, it is difficult to interpret the findings. 

* Minor: Spell out abbreviations when used for the first time (e.g., Page 3 WISE intervention)

[LINK]

---

## [Decision Letter · Decision Letter 2]

13 Sep 2021

Dear Dr. Kidger,

Thank you very much for re-submitting your manuscript "An intervention to improve teacher wellbeing support and training to support students in UK high schools (the WISE study): a cluster randomised controlled trial" (PMEDICINE-D-21-02018R2) for review by PLOS Medicine.

I have discussed the paper with my colleagues and the academic editor and it was also seen again by three reviewers. I am pleased to say that provided the remaining editorial and production issues are dealt with we are planning to accept the paper for publication in the journal.

[LINK]

We look forward to receiving the revised manuscript by Sep 20 2021 11:59PM.   

Sincerely,

Beryne Odeny, 

Associate Editor 

PLOS Medicine

plosmedicine.org

Requests from Editors:

1) In the footnotes of tables (and figures) please include definitions for all abbreviations e.g., OR, IQR, SD, WEMWBS, PHQ-8

2) References: please ensure that journal name abbreviations match those found in the National Center for Biotechnology Information (NCBI) databases and are appropriately formatted and capitalized. If using a reference manager, please select the “PLOS Medicine” citation style.

3) The terms gender and sex are not interchangeable (as discussed in http://www.who.int/gender/whatisgender/en/ ); please use the appropriate term.

Comments from Reviewers:

Reviewer #2: Thank you to the authors for addressing all my previous comments well.

Reviewer #3: The Authors have taken care of all the suggested changes. Great work. I therefore recommend publication of this manuscript.

Reviewer #5: I congratulate the authors for adequately responding to the comments. I have no further comments.

[LINK]

---

## [Editor Report · Decision Letter 3]

8 Oct 2021

Dear Dr. Kidger,

Thank you very much for re-submitting your manuscript "An intervention to improve teacher wellbeing support and training to support students in UK high schools (the WISE study): a cluster randomised controlled trial" (PMEDICINE-D-21-02018R3) for review by PLOS Medicine.

I have discussed the paper with my colleagues and the academic editor. I am pleased to say that provided the remaining editorial and production issues are dealt with we are planning to accept the paper for publication in the journal.

The one remaining issue that needs to be addressed is listed at the end of this email:

[LINK]

We look forward to receiving the revised manuscript by Oct 15 2021 11:59PM.   

Sincerely,

Beryne Odeny, 

PLOS Medicine

plosmedicine.org

Requests from Editors:

Thank for responding to prior editorial requests. Before we proceed, please address the following comment:

1) Please avoid assertions of primacy such as "We conducted the first ever cluster randomised ..." and say "To our knowledge, we conducted the first ever ..."

Comments from Reviewers:

[LINK]

---

## [Editor Report · Decision Letter 4]

12 Oct 2021

Dear Dr Kidger, 

On behalf of my colleagues and the Academic Editor, Dr. Mark Tomlinson, I am pleased to inform you that we have agreed to publish your manuscript "An intervention to improve teacher wellbeing support and training to support students in UK high schools (the WISE study): a cluster randomised controlled trial" (PMEDICINE-D-21-02018R4) in PLOS Medicine.

PRESS

Sincerely, 

Beryne Odeny 

PLOS Medicine